# Exploring the non-communicable disease (NCD) network of multi-morbid individuals in India: A network analysis

**Parul Puri** *, **Shri Kant Singh**

Department of Survey Research and Data Analytics, International Institute for Population Sciences, Mumbai, Maharashtra, India

* parulpuri93@gmail.com

## Abstract

Nationally representative evidence discussing the interplay of non-communicable diseases (diseases) are scarce in India. Therefore, the present study aims to fill this research void by providing empirical evidence on disease networking using a large nationally representative cross-sectional sample segregated by gender among older adults in India. The analysis utilized data on 10,606 multimorbid women and 7,912 multimorbid men from the Longitudinal Ageing Study in India (LASI), 2017–18. Multimorbidity was defined as the co-occurrence of two or more diseases in an individual using a list of 16 self-reported diseases. Weighted networks were visualized to illustrates the complex relationships between the diseases using network analysis. The findings suggest that women possess a higher burden of multimorbidity than men. Hypertension, musculoskeletal disorder, gastrointestinal disorder, diabetes mellitus, and skin diseases were reported as the most recurrent diseases. 'Hypertension-musculoskeletal disorder', 'diabetes mellitus-hypertension', 'gastrointestinal disorders-hypertension' and 'gastrointestinal disorders- musculoskeletal disorder' were recurrent disease combinations among the multimorbid individuals. The study generated compelling evidence to establish that there are statistically significant differences between the prevalence of diseases and how they interact with each other between women and men. These findings further accentuate that disease networks are slightly more complex among women. In totality, the study visualizes disease association, identifies the most influential diseases to the network, and those which acts as a bridge between other diseases, causing multimorbidity among the older adult population in India.

## Introduction

Epidemiological transition coupled with lifestyle and behavioral modifications has synergistically accelerated the global non-communicable disease (NCD) burden [1]. The World Health Organization (WHO) suggested that in 2016, 71 percent of global deaths were contributed by NCDs, and 77 percent of all NCD deaths occurred in low-and-middle-income countries (LMICs) [1–3]. This rising NCD burden accompanied with lifestyle and behavioral

**Data Availability Statement:** The data has been archived in the public repository of LASI held at IIPS. The access to the data requires registration which is granted specifically for research purposes. The study utilised a de-identified data from a

secondary data source, Longitudinal Ageing Study in India, 2017-18. The survey followed all necessary guidelines and received ethical approval from the Indian Council of Medical Research (ICMR). Additional approval on survey protocols was provided by the Institutional Review Board (IRB) held at IIPS, Mumbai India. At the unit level, individuals were supplied with a catalogue containing the information on the purpose of the survey, confidentiality, and safety of health assessment. Written consent forms were administered at household and individual levels, in accordance with the Human Subject Protection. The data can be accessed using the link: https://www.iipsindia.ac.in/content/lasi-wave-i. Access to the dataset requires registration and is granted only for legitimate research purposes. The application to access dataset is available at https://iipsindia.ac.in/sites/default/files/LASI_DataRequestForm_0.pdf. A duly-filled form has to be emailed to Information, Communication and Technology (ICT) Unit held at the International Institute for Population Sciences, Mumbai at datacenter@iipsindia.ac.in.

**Funding:** The authors received no specific funding for this work.

**Competing interests:** The authors have declared that no competing interests exist.

modifications has resulted in simultaneous occurrence of multiple health conditions in an individual, alias multimorbidity [4,5]. Multimorbidity, usually defined as the simultaneous occurrence of two or more chronic (long-term) conditions, is becoming fairly common in LMICs, with around 30 percent of the population affected with it [5,6].

India is no exception to the existing norm and is experiencing the burden of multimorbidity from past two decades [7]. Findings from Longitudinal Ageing Study in India, suggests that 18 percent of the individuals aged 45 years or older were affected with multimorbidity in 2018 [8]. Studies have highlighted linkages of multimorbidity with high healthcare utilization and expenditure [9], poor quality of life [10,11], low self-rated health [11,12], increased frailty [13], disability [11,12,14] and mortality [6]; this makes it a serious public health concern for the government of India [15]. Despite the deleterious implications the need for effective management of multimorbidity was given its due attention post COVID-19 pandemic [16]. However, till date the empirical shreds of evidence on multimorbidity remain inadequate at the national level [5,17,18]. Existing literature draws conclusions on the basis of the smaller sub-samples that are drawn on selected sub-groups of the population, primarily located at healthcare facilities [5,17–21]. In addition, the studies providing nationally representative evidence are based on lesser number of diseases and employ crude measures like chronic disease score to operationally define multimorbidity [7,19,22]. In totality, there is a dearth in the studies exploring the interplay between disease and thus, they are unable to encapsulate the linkages between diseases among the multimorbid population in the country.

The present study aims to decode the complexities of multiple non-communicable diseases among older adults (individuals aged 45 years or older) in India. The study provides empirical evidence on disease networking using a large nationally-representative sample on older adults. The study presents an in-depth comparative analysis, which provides an overview on most recurrent diseases and dyads (disease combinations). Furthermore, the study visualizes the disease networks and gives insights on the diseases which are most prominent (influential) to the network and which acts as a bridge between two diseases; and thus, can be treated as a precursor to multimorbidity.

## Materials and methods

### Data source and sampling design

The data employed for the present analysis is obtained from the Longitudinal Ageing Study in India (LASI), wave-1, 2017–18. LASI was conducted under the stewardship of the Ministry of Health and Family Welfare, Government of India [8]. LASI implemented a multi-stage stratified probability cluster sampling design to draw nationally representative data from 35 states/union territories (except Sikkim). For each state, LASI stratified the sampling design on the basis of residence, i.e., rural and urban. A three-stage sampling design was adopted for rural areas, whereas a four-stage sampling design was adopted for the urban areas.

In rural areas, stage one involved selection of a Primary Sampling Unit (PSU), this included selection of sub-districts (Tehsils/Talukas). In the second stage, a Secondary Sampling Unit (SSU) was selected, thus included selection of village. In the third stage, households were selected, from which eligible respondents were interviewed. Whereas in urban areas, stage one involved selection of a PSU, this included selection of sub-districts (Tehsils/Talukas). In the second stage, a SSU was selected, i.e., urban wards. Once an urban ward is selected a Census Enumeration Block (CEB) was selected in the third stage. Following which households were selected in the fourth stage.

For the selection of PSU, a sampling frame was chosen from the 2011 census. In rural areas, the sampling frame in the second stage was the villages in all selected sub-districts. The list of

CEBs in each selected ward was the sampling frame in the third stage. To obtain the sampling frame for the selection of households from secondary sampling units (SSUs), a mapping and household listing operation was carried out in the sampled SSUs (i.e., villages in rural areas and CEBs in urban areas). All of the listed households in selected villages/CEBs formed the sampling frame for the selection of households. A detailed account of the survey design, sampling frame and sample size can be seen elsewhere (8).

### Ethical consideration

LASI received ethical approval from the Indian Council of Medical Research (ICMR) and Institutional Review Board held at International Institute for Population Sciences (IIPS), India. During the fieldwork, a catalogue containing the information on the purpose of the survey, confidentiality and safety of health assessment was provided to each eligible participant. In addition, separate written consent forms were administered at household and individual levels, in accordance with the Human Subject Protection [8]. There was no personal information included in LASI, and therefore ethical approval for open use of LASI data was not required.

### Study population

LASI contained de-identified data on 72,250 individuals above the age of 45 years and their spouses irrespective of age. From the retrieved dataset, information on 65,562 individuals above the age of 45 years were derived. Further, employing the operational definition multimorbidity, i.e., simultaneous occurrence of two or more chronic diseases, 18,518 multimorbid individuals (to be referred as individuals from now onwards) were identified. These 18,518 individuals included 10,606 multimorbid women (to be referred as women from now onwards) and 7,912 multimorbid men (to be referred as men from now onwards). A description of the sample selection is presented in Fig 1.

### Measures

LASI commissioned the question, "Has any health professional ever diagnosed you with the following chronic conditions or diseases?". The study included a list of 16 non-communicable diseases, namely asthma (AS), cancer (CA), chronic bronchitis (CB), chronic heart disease (CHD), chronic obstructive pulmonary disease (COPD), chronic renal failure (CRF), diabetes

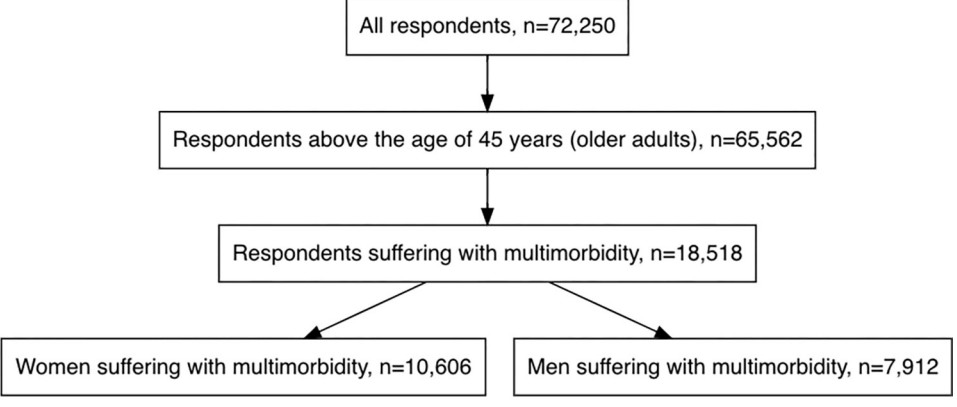

**Fig 1. Description of study sample, Longitudinal Aging Study in India, 2017–18.**

mellitus (DM), gastrointestinal disorder (GD), high cholesterol (HC), hypertension (HYP), musculoskeletal disorder (MKS), neurological and psychiatric disorder (NPD), skin disease (SD), stroke (ST), thyroid disease (THY) and urinary incontinence (UI). All the aforementioned diseases were self-reported medical diagnoses and were classified into binary form: absent- '0' and present- '1'. Owing to the social and biological differences between women and men, the current analysis was segregated by the gender of the individual.

## Statistical analysis

**Preliminary analysis.** The prevalence of selected diseases among the multimorbid population was reported. These findings were supplemented with chi-square p-values to identify the statistically significant difference between the disease burden among women and men. From the sixteen diseases included in the analysis, a total of $^{16}C_2 = 120$ combinations (dyads) were explored; heat maps were generated to identify recurrent dyads for both women and men.

**Network analysis.** The main statistical procedure employed in the study was a network analysis. A network essentially comprises of two components, namely nodes and edges. All sixteen diseases were represented by nodes (represented by circles), whereas edges (━) represented the association between two diseases. For each of the two components two separate input files were generated. Nodes file contained the information on disease prevalence, whereas edges file included information on disease associations. For constructing the edges file, all possible disease combinations were utilized, an array of age-adjusted binary logistic regression models were executed. Age-adjusted odds ratios along with p-values were reported for each dyad combination. Any specific association (dyad combination) was included in the final edge file, if the conditions below are satisfied:

i. All the associations' which are statistically significant i.e., p-value<0.05 [23].

ii. For relationships which satisfy point (i), the age-adjusted odds ratio should have a value greater or equal to 1.2, i.e., OR ≥1.2 (23).

This was done to ensure that only statistically significant positive associations are selected for the final disease network. The size of the node represented the prevalence of the disease, whereas the edge thickness was proportional to the degree of association (age-adjusted odds ratio) between two diseases. Thus, bigger the node, higher the prevalence of the NCD, and thicker the edge, stronger is the association between the two diseases. The information from nodes and final edges file was utilized to create an edgelist, which served as an input to visualize the final disease network. The final disease network was a weighted undirected network as all the edges included in the study were of bi-directional nature.

**Network and node attributes.** For the full network, four attributes were computed, namely number of nodes, number of edges, network diameter and network density. Number of nodes refers to the total number of network units which in present case is NCDs [24]. Whereas, the number of edges is the number of links (statistically significant and positive associations) connecting the nodes (NCDs) [24]. More number of nodes refers to a higher number of statistically significant and positive association, which depicts more complex disease networks. Network diameter refers to the maximum distance between any two nodes (diseases) in the network. It is an indicator of network cohesiveness, i.e., how united is a network. Diameter ranges between zero and one, where zero indicates no incohesive, whereas one indicates complete cohesiveness [23]. While network density measure the sparsity, and is defined as the number of actual connections divided by the potential connections in any network [23,24]. Higher network density represents, more complicated disease networks. In addition, positional features were computed for individual nodes (diseases). These include measures of

centrality including degree (local measure), node closeness centrality (global measure) and node betweenness centrality were reported.

Degree ($K_i$) measures the connectivity of a node 'i'. It represents how involved a specific node is in a network [23]. It is defined as:

$$K_i = n_{i_{in}} + n_{i_{out}}$$

Where,

$K_i$ is the degree for node 'i'

$n_{i_{in}}$ is the number of ties directed inwards and

$n_{i_{out}}$ number of ties directed outwards

Dense networks are highly connected networks where NCDs can quickly move across the networks. Therefore, denser the network, easier is the disease propagation. Node closeness centrality (or closeness centrality) represents the connectedness of a disease (node) to a network. It is a score (sum) which is constructed for each disease (node) on their closeness (shortest path) to all other nodes within the network. Higher the closeness centrality, more important position it holds in the disease network. In other words, that specific disease is connected with a greater number of NCDs in the networks [23,25,26].

Node betweenness centrality (or betweenness) represents the potential influence of a node (disease) on the distribution of other nodes (diseases). It is defined as the number of times a particular node (diseases) acted as a link in the shortest paths between other nodes (diseases) [23,25]. Thus, a disease with higher betweenness centrality means that it acts as a bridge between other diseases through common pathophysiological mechanism or shared risk factors. Thus, such diseases hold utmost importance in forming these networks.

All the estimates generated in the study were presented after suitable application of sampling weights provided by LASI, 2017–18 [8]. Analysis for identifying and visualizing disease network was conducted using in RStudio version 1.1.463 (R Studio, Inc.). The study followed the Strengthening the Reporting of Observational Studies in Epidemiology (STROBE) reporting guidelines for cross-sectional studies presented in S1 Table.

## Results

### Burden of individual non-communicable diseases among multimorbid older adults

The study included 18,518 multimorbid individuals, which included 10,606 (56.9%) women and 7,912 (43.1%) men. Table 1 provides the prevalence of selected chronic non-communicable diseases among the multimorbid population. The findings suggest that burden of asthma (p-value = 0.000), cancer (p-value = 0.005), chronic heart disease (p-value = 0.000), coronary obstructive pulmonary disease (p-value = 0.000), chronic renal failure (p-value = 0.000), diabetes mellitus (p-value = 0.000), hypertension (p-value = 0.000), musculoskeletal disorder (p-value = 0.000), neurological and psychiatric disorder (p-value = 0.002), skin disease (p-value = 0.000), stroke (p-value = 0.000), and thyroid disease (p-value = 0.000) were significantly different between women and men.

For women, hypertension (71.7%), musculoskeletal disorder (44.7%), gastrointestinal disorder (38.3%), diabetes mellitus (34.7%), and skin disease (11.9%) were five most prominently occurring NCDs, which is similar to that reported by the total population. For men, however, the diseases remained same, but their ranking was modified, i.e., hypertension (62.8%), gastrointestinal disorder (39.3%), diabetes mellitus (35.9%), musculoskeletal disorder (32.4%), and skin disease (15.0%) were most commonly occurring diseases.

**Table 1. Prevalence of selected chronic morbidities in the population stratified by gender, longitudinal study in India, 2017–18.**

| Non-communicable Diseases | Total (n = 18518) | Men (n = 7912) | Women (n = 10606) | p-value |
|---|---|---|---|---|
| | n (%) | n (%) | n (%) | |
| Asthma (AS) | 1838 (12.28) | 909 (14.30) | 929 (10.75) | 0.000 |
| Cancer (CA) | 317 (1.64) | 111 (1.26) | 206 (1.93) | 0.005 |
| Chronic Bronchitis (CB) | 477 (3.47) | 216 (2.94) | 261 (3.87) | 0.253 |
| Chronic Heart Disease (CHD) | 2028 (11.46) | 1063 (12.96) | 965 (10.33) | 0.000 |
| Chronic Obstructive Pulmonary Disease (COPD) | 587 (4.05) | 303 (3.73) | 284 (4.28) | 0.000 |
| Chronic Renal Failure (CRF) | 395 (1.94) | 216 (2.54) | 179 (1.49) | 0.000 |
| Diabetes (DM) | 6716 (35.23) | 3071 (35.94) | 3645 (34.69) | 0.000 |
| Gastrointestinal Disorder (GD) | 7546 (38.74) | 3217 (39.27) | 4329 (38.34) | 0.830 |
| High Cholesterol (HC) | 2119 (7.01) | 882 (7.28) | 1237 (6.81) | 0.276 |
| Hypertension (HYP) | 12895 (67.85) | 5187 (62.80) | 7708 (71.68) | 0.000 |
| Musculoskeletal Disorder (MKS) | 6655 (39.38) | 2305 (32.41) | 4350 (44.67) | 0.000 |
| Neurological and Psychiatric Disorder (NPD) | 1160 (6.81) | 545 (7.20) | 615 (6.51) | 0.002 |
| Skin Diseases (SD) | 2245 (12.83) | 1050 (14.99) | 1195 (11.91) | 0.000 |
| Stroke (ST) | 1008 (5.63) | 604 (7.91) | 404 (3.91) | 0.000 |
| Thyroid Disease (TD) | 1577 (8.10) | 295 (3.82) | 1282 (11.34) | 0.000 |
| Urinary Incontinence (UI) | 1470 (8.59) | 642 (9.05) | 828 (8.23) | 0.444 |

## Non-communicable disease combinations in multimorbid older adult population

Fig 2 represents two heat maps for multimorbid women and men, respectively. Heat maps depict the prevalence of all possible disease combinations (120 combinations for women and men each) among the multimorbid older adult population.

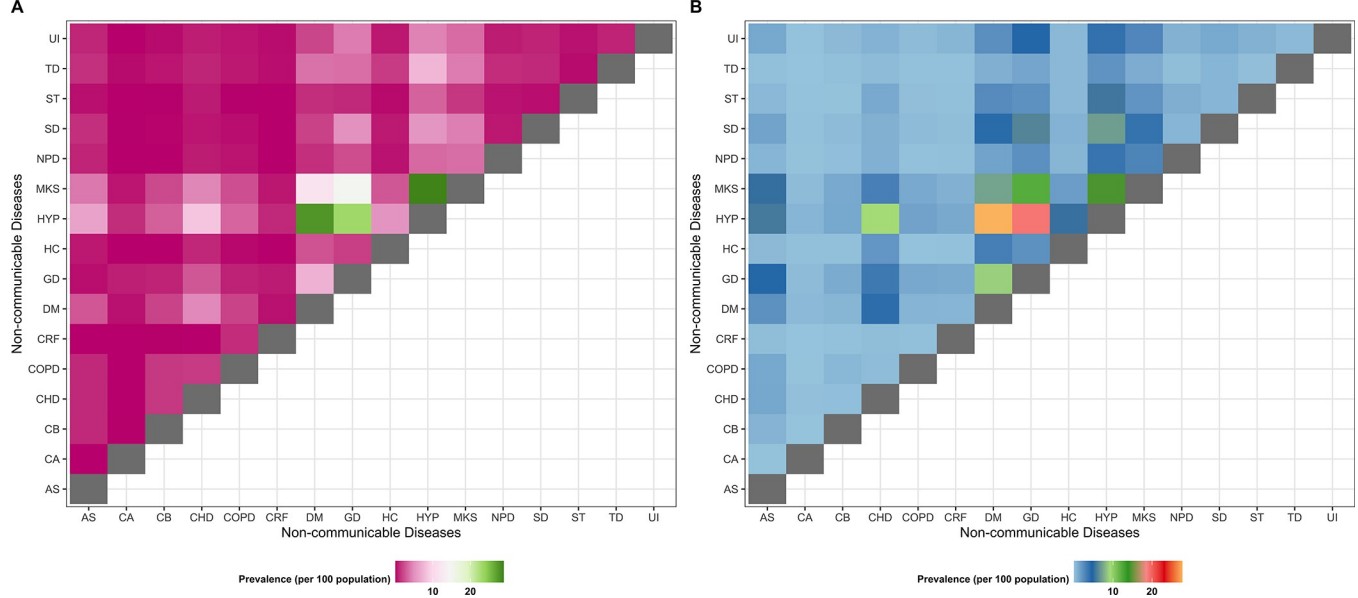

**Fig 2.** Heat maps depicting non-communicable disease dyads prevalence for (A) women and (B) men aged 45 years and above, Longitudinal Ageing Study in India (LASI), 2017–18.

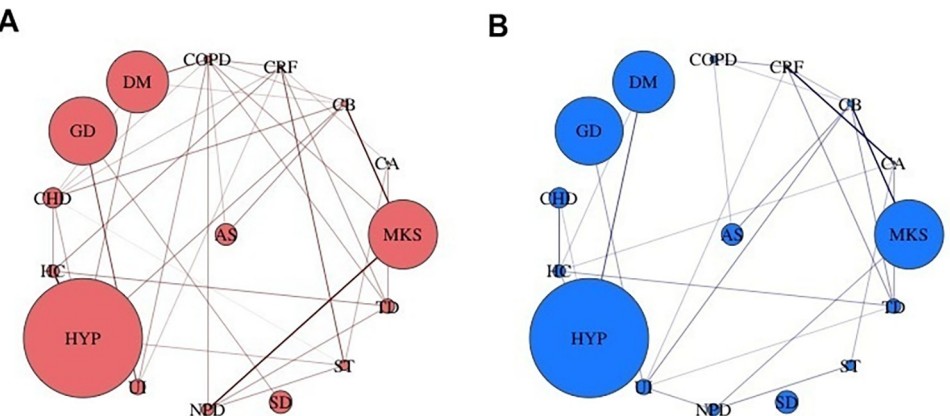

**Fig 3. Multimorbidity networks for (A) women and (B) men aged 45 years and above, considering inclusion criterion for sixteen non-communicable diseases, Longitudinal Ageing Study in India (LASI), 2017–18.**

Among women, 55 out of the 120 combinations had a prevalence greater than one percent. 'Hypertension-musculoskeletal disorder' (28.9%), 'diabetes mellitus-hypertension' (27.8%), 'gastrointestinal disorders-hypertension' (23.2%), 'gastrointestinal disorders- musculoskeletal disorder' (14.9%) and 'diabetes mellitus- musculoskeletal disorder' (11.5%) were five most recurrent disease combinations among the multimorbid women. Whereas, in men 52 out of the 120 combinations had a prevalence greater than one percent. 'Diabetes mellitus-hypertension' (27.8%), 'gastrointestinal disorders-hypertension' (19.1%), 'hypertension-musculoskeletal disorder' (14.5%), 'gastrointestinal disorders- musculoskeletal disorder' (12.2%) and 'chronic heart disease- musculoskeletal disorder' (9.3%) were five most recurrent disease combinations among the multimorbid men. Prevalence of all possible dyad combinations by gender can be seen in S2 Table.

## Non-communicable disease network among older adults

Fig 3 represents undirected network with positively and statistically significantly associated diseases (p-value<0.05, OR ≥1.2). Comparing the disease networks for women and men, both had equal number of nodes i.e., 16 diseases (network units). Initially, the total number of edges i.e., the number of disease association explored were 120 for each network, however, to ensure that only statistically significant positive associations are visualized, two inclusion criteria were employed, after which 35 and 25 associations were remaining among women and men, respectively (Table 2). Age-adjusted association (OR values) for all possible disease pairs by gender is presented in S3 Table. Women network had a greater diameter as compared to men; while both networks were equally sparse. In addition, full multimorbidity networks can be seen in S1 Fig.

**Table 2. Network specific attributes by gender in the older adult population with multimorbidity, Longitudinal Ageing Study in India, 2017–18.**

| Network Attributes | Disease Network | |
|---|---|---|
| | Women | Men |
| Number of nodes | 16 | 16 |
| Number of edges (OR ≥1.2, p-value<0.05) | 35 | 25 |
| Diameter | 2.21 | 1.04 |
| Density | 0.50 | 0.50 |

**Table 3. Node specific centrality measures by gender in the older adult population with multimorbidity, Longitudinal Ageing Study in India, 2017–18.**

| Non-Communicable Diseases (NCDs) | Degree | | Closeness Centrality (in %) | | Betweenness Centrality (in %) | |
|---|---|---|---|---|---|---|
| | Women | Men | Women | Men | Women | Men |
| Asthma (AS) | 2 | 2 | 1.50 | 1.19 | 0.00 | 0.00 |
| Cancer (CA) | 3 | 4 | 1.80 | 1.60 | 1.00 | 9.00 |
| Chronic Bronchitis (CB) | 6 | 6 | 1.91 | 1.60 | 5.00 | 20.00 |
| Chronic Heart Disease (CHD) | 7 | 2 | 2.00 | 0.98 | 3.00 | 0.00 |
| Coronary Obstructive Pulmonary Disease (COPD) | 9 | 3 | 2.66 | 1.13 | 44.00 | 0.00 |
| Chronic Renal Failure (CRF) | 7 | 5 | 2.12 | 1.52 | 8.00 | 12.00 |
| Diabetes (DM) | 4 | 2 | 2.18 | 1.20 | 6.00 | 0.00 |
| Gastrointestinal Disorder (GD) | 2 | 1 | 1.41 | 1.22 | 14.00 | 0.00 |
| High Cholesterol (HC) | 4 | 5 | 1.80 | 1.53 | 3.00 | 33.00 |
| Hypertension (HYP) | 5 | 3 | 1.91 | 1.20 | 10.00 | 1.00 |
| Musculoskeletal Disorder (MKS) | 3 | 2 | 2.03 | 1.34 | 6.00 | 0.00 |
| Neurological and Psychiatric Disorder (NPD) | 4 | 3 | 2.11 | 1.38 | 6.00 | 5.00 |
| Skin Diseases (SD) | 1 | 0 | 1.12 | 0.42 | 0.00 | 0.00 |
| Stroke (ST) | 5 | 2 | 1.99 | 1.39 | 6.00 | 1.00 |
| Thyroid Disease (TD) | 5 | 5 | 2.21 | 1.83 | 9.50 | 45.00 |
| Urinary Incontinence (UI) | 3 | 5 | 1.97 | 1.61 | 26.00 | 22.00 |

Table 3 provides node specific centrality measures. Among women, the diseases with more connections in the networks were coronary obstructive pulmonary disease (9), chronic heart disease (7), chronic renal failure (7), chronic bronchitis (6), stroke (5), thyroid disease (5), and hypertension (5). Considering the closeness centrality, coronary obstructive pulmonary disease (2.7%), thyroid disease (2.2%), diabetes mellitus (2.2%), chronic renal failure (2.1%), and neurological and psychiatric disorder (2.1%) were placed at the best positions to impact the whole disease network quickest. Betweenness, which represents the number of times a specific node (disease) acted as a link between the shortest path between two other diseases; coronary obstructive pulmonary disease (44.0%), urinary incontinence (26.0%), gastrointestinal disorder (14.0%), hypertension (10.0%), and thyroid disorder (9.5%) acted a bridge between two diseases.

Among men, the diseases with more connections in the networks were chronic bronchitis (6), chronic renal failure (5), high cholesterol (5), thyroid disease (5), urinary incontinence (5), and cancer (4). Considering the closeness centrality, thyroid disease (1.8%), urinary incontinence (1.6%), chronic bronchitis (1.6%), cancer (1.6%), and high cholesterol (1.5%) were placed at the best positions to impact the whole disease network quickest. Thyroid diseases (45.0%), high cholesterol (33.0%), urinary incontinence (22.0%), chronic bronchitis (20.0%), and chronic renal failure (12.0%) acted a bridge between two disease highest number of times.

## Discussion

The present study provides empirical evidence using a nationally representative data on 18,518 multimorbid older individuals (10,606 women and 7,912 men); from the first wave of the Longitudinal Ageing Study in India (LASI), 2017–18. The findings suggest that women (56.9%) possess a higher burden of multimorbidity than men (43.1%) in India. Existing literature in India, have reported a significantly higher burden of multimorbidity among women [7,27] as compared to men. The primary reasons for higher disease burden is increased life expectancy,

which enables them to encounter more number of vital events in life, which have a direct or indirect relationship with expanding chronic disease burden [27].

Hypertension, musculoskeletal disorder, gastrointestinal disorder, and diabetes mellitus were reported as most recurrent NCDs for multimorbid women and men. These findings are in concordance with the studies based in South Asian countries, which suggest a preponderance hypertension [7,20,25,27], musculoskeletal disorders [20,27], gastrointestinal disorder [20,25,27], and diabetes [7,20,25]. 'Hypertension-musculoskeletal disorder' and 'diabetes mellitus-hypertension' were recurrent disease combinations among the multimorbid women and men. Musculoskeletal disorders comprise of a wide range of diseases and conditions, most common include tendonitis, osteoarthritis and rheumatoid arthritis. While some studies report a negative association between osteoarthritis (a musculoskeletal disorder) and hypertension [28,29]; others report a positive association between hypertension and musculoskeletal complaints, like rheumatoid arthritis and musculoskeletal pain [30–32]. These studies suggest that musculoskeletal pain affect the Autonomic Nervous System (ANS) that controls the cardiovascular activities and regulates blood pressure levels and heart rate, and this might interfere with both diastolic and systolic blood pressure levels [31]. On the other hand, diabetes mellitus and hypertension are stated to have bi-directional relationship with shared risk factors. Also, literature suggest that diabetes can intensify age-specific blood pressure dysfunction [21,33].

The findings further emphasize that linkages between the diseases are much more complex in women as compared to men. This is because, on the basis of the inclusion criterion applied, a greater number of associations were remaining in women. In absence of comparable studies in India, we refer to evidence generated by Schäfer et al. (2014), which represents similar findings [25]. The study generated compelling evidence to establish that there are statistically significant differences between the prevalence of diseases and in the way, they interact with each other for women and men. For instance, in women, coronary obstructive pulmonary disease, thyroid disease, and diabetes mellitus were placed at the best positions to impact the whole disease network quickest; whereas, coronary obstructive pulmonary disease, along with urinary incontinence, and gastrointestinal disorder acted a bridge between two diseases majority of the times. In men however, thyroid disease was placed at the best positions to impact the whole disease network; whereas, thyroid disease and high cholesterol acted a bridge between two disease majority of the times.

## Strengths and limitations

The strength of our study lies in the use of a recently published large-scale nationally representative data on older adults in India. The study uses a list of sixteen NCDs (diseases), and employed a globally accepted definition of 'simultaneous occurrence of two or more chronic diseases' to measure multimorbidity. In addition, the use of network analysis, has provided a novel perspective on diseases interaction in India, which by far has been overlooked. The study utilized a rigorous inclusion criterion for selecting statistically significant positive associations which provides robustness to the findings. All the disease associations depicted in the study are sex-specific and are adjusted for age of the multimorbid individual.

The diseases included in the study are self-reported, which might lead to misclassification bias. Directed networks were not studied in the present analysis as age at diagnoses was not asked for all the diseases included in the study. To assess the generalizability of our study findings, it is essential to replicate our study using different data sets (if available) for older adult population in India.

### Implications of findings

Although in the strict medical viewpoint, NCDs are non-transmissible between individuals. However, owing to the deep-rooted linkages NCDs share with geographical and behavioral aspects, like temperature, altitude, precipitation, pollution levels, dietary influences, lifestyle modifications, exposure to risky health behaviors, non-compliance with medical regimes and avoidance towards health care [34]; it would not be an exaggeration to affirm that, each individual has the tendency and capacity to alter the behaviors of another individual, living in the vicinity; which, can be referred to as "neighborhood affect".

Considering the aforementioned explanation, which acts as a catalyst for diseases with shared pathophysiologies and risk factors, NCDs are not rigorously non-transmissible. Despite this, existing evidence on multimorbidity in India, has by far been following a unidirectional approach, i.e., chronic disease score (CDS)-based approach [4,5,10], which fails to study disease linkages, and hence miss out vital information, which can be used in devising community-oriented treatment and management regimens.

Considering the social and biological differences, our study decodes this vital piece of information for the older women and men in India. The study visualizes the complicated relationships between diseases, and identifies the NCDs that are most influential to the network, and those which acts as a bridge between diseases. These findings can assist physicians in understanding the interplay between diseases and can be used for mending existing treatment strategies, reducing the likelihood of multimorbidity-related organ failure and polypharmacy. These findings can further be employed for the purpose of drug repurposing in India.

## Conclusion

Our study provides empirical evidence which establishes that multimorbidity is an emergent public health concern among older women and men in India. The findings indicate that issue of multiple morbidities (diseases) is much more complicated and is not yet decrypted in its full extent. The findings highlight most recurrent diseases, frequently occurring disease combinations, and their association among older adults in India. These findings may assist policymakers in drafting and issuing improved guidelines to effectively and adequately manage the chronic non-communicable disease progression and thus, combat the escalating burden of multimorbidity among older adults in India.

## Supporting information

**S1 Fig. Full multimorbidity networks for (A) women and (B) men aged 45 years and above for sixteen non-communicable diseases, LASI, 2017–18.**
(PDF)

**S1 Table. STROBE statement—checklist of items that should be included in reports of cross-sectional studies.**
(PDF)

**S2 Table. Prevalence of all possible dyad combinations by gender among older adults in India, LASI, 2017–18.**
(PDF)

**S3 Table. Age-adjusted association between selected non-communicable diseases (NCDs) segregated by gender, among older adults in India, LASI, 2017–18.**
(PDF)

## Acknowledgments

The Longitudinal Ageing Study in India (LASI) is responsible for assembling and publishing accurate, nationally representative data on the key indicators of physical, mental, psychological, and social health and well-being of the rapidly aging Indian population. LASI was partnered by the International Institute for Population Sciences (IIPS), Harvard T. H. Chan School of Public Health, and the University of Southern California; and was launched under the tutelage of the Ministry of Health and Family Welfare Government of India. The authors would like to acknowledge all the research coordinators who developed the study's research protocol. This work has been presented at the IUSSP International Population Conference 2021.

## Author Contributions

**Conceptualization:** Parul Puri.

**Data curation:** Parul Puri.

**Formal analysis:** Parul Puri.

**Investigation:** Parul Puri.

**Methodology:** Parul Puri.

**Software:** Parul Puri.

**Supervision:** Shri Kant Singh.

**Visualization:** Parul Puri.

**Writing – original draft:** Parul Puri.

**Writing – review & editing:** Parul Puri, Shri Kant Singh.

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
