## [Decision Letter · Decision Letter 0]

16 Mar 2022

PGPH-D-21-00569

Exploring the non-communicable disease (NCD) network of multi-morbid individuals in India: A network analysis

Dear Dr.Puri,

Thank you for submitting your manuscript to PLOS Global Public Health. After careful consideration, we feel that it has merit but does not fully meet PLOS Global Public Health’s publication criteria as it currently stands. Therefore, we invite you to submit a revised version of the manuscript that addresses the points raised during the review process.

As the manuscript uses complex statistical methods, you may consider explaining the methods and interpretation of results with explanations and clarity to make the manuscript interesting and useful for wider audiences.

We look forward to receiving your revised manuscript.

Kind regards,

Prabhdeep Kaur, DNB Medicine, MAE (Epidemiology)

Academic Editor

Journal Requirements:

1. Please amend your Financial Disclosure statement. If you did not receive any funding for this study, please simply state: “The authors received no specific funding for this work.”

2. Please update your Competing Interests statement. If you have no competing interests to declare, please state: “The authors have declared that no competing interests exist.”

3. Please note that your Data Availability Statement is currently missing a direct link to access each database. If your manuscript is accepted for publication, you will be asked to provide these details on a very short timeline. We therefore suggest that you provide this information now, though we will not hold up the peer review process if you are unable.

4. Please provide separate figure files in .tif or .eps format only and ensure that all files are under our size limit of 20MB.

5. Please include a copy of Table 1 which you refer to in your text on page 8.

Additional Editor Comments (if provided):

Reviewers' comments:

Reviewer's Responses to Questions

**Comments to the Author**

1. Does this manuscript meet PLOS Global Public Health’s publication criteria? Is the manuscript technically sound, and do the data support the conclusions? The manuscript must describe methodologically and ethically rigorous research with conclusions that are appropriately drawn based on the data presented.

Reviewer #1: Yes

Reviewer #2: Partly

2. Has the statistical analysis been performed appropriately and rigorously?

Reviewer #1: Yes

Reviewer #2: Yes

3. Have the authors made all data underlying the findings in their manuscript fully available (please refer to the Data Availability Statement at the start of the manuscript PDF file)?

Reviewer #1: Yes

Reviewer #2: No

4. Is the manuscript presented in an intelligible fashion and written in standard English?

Reviewer #1: Yes

Reviewer #2: Yes

5. Review Comments to the Author

Reviewer #1: It is very interesting and important one on Noncommunicable Diseases. Specially the authors try to analyzed the multi-morbidity of NCDs among the patients. Particularly, multi-morbidity always make the disease progression more complex and quality of life become poor. To prevention of multi-morbidity we need to know first about the pattern of it. 

Reviewer #2: Puri and colleagues aimed to decode the complexities of multiple non-communicable diseases

among older adults (individuals aged 45 years or older) in India using network analysis. The manuscript has merit and definitely important considering the novel analysis techniques which the authors used. However, authors should address the following comments in order to improve the quality of the manuscript.

Major Comments:

1. Please provide description of the data source and the geographic area it covered.

2. Data collection process including sampling, sampling frame, tools and questionnaires should be described.

3. The first and foremost limitations of the study is the multimorbidity’s were self-reported. A description regarding the data collection process and how the participants were asked about the multimorbidity should be added.

4. Explicit explanation of the following information needed in the methods section:

• Number of nodes

• Number of edges

• Diameter

• Density

What is the significance of the number of these elements? If the number is higher, what does it imply? If the number is lower, what does it imply?

5. Similarly, what is the implication of Degree, Closeness Centrality (in %), Betweenness Centrality (in %)?

6. In the first line of discussion section, the authors mentioned that it is a nationally representative sample. Did the authors adjust the weighting and cluster effect during analysis?

7. Please highlight policy level implications of the research.

6. PLOS authors have the option to publish the peer review history of their article (what does this mean?). If published, this will include your full peer review and any attached files.

**Do you want your identity to be public for this peer review?** For information about this choice, including consent withdrawal, please see our Privacy Policy.

Reviewer #1: **Yes: **Palash Chandra Banik

Reviewer #2: **Yes: **Rajat Das Gupta

---

## [Decision Letter · Decision Letter 1]

9 Jun 2022

Exploring the non-communicable disease (NCD) network of multi-morbid individuals in India: A network analysis

PGPH-D-21-00569R1

Dear Parul Puri

We are pleased to inform you that your manuscript 'Exploring the non-communicable disease (NCD) network of multi-morbid individuals in India: A network analysis' has been provisionally accepted for publication in PLOS Global Public Health.

Best regards,

Prabhdeep Kaur, DNB Medicine, MAE (Epidemiology)

Academic Editor

Reviewer Comments (if any, and for reference):

Reviewer's Responses to Questions

**Comments to the Author**

1. If the authors have adequately addressed your comments raised in a previous round of review and you feel that this manuscript is now acceptable for publication, you may indicate that here to bypass the “Comments to the Author” section, enter your conflict of interest statement in the “Confidential to Editor” section, and submit your "Accept" recommendation.

Reviewer #2: All comments have been addressed

2. Does this manuscript meet PLOS Global Public Health’s publication criteria? Is the manuscript technically sound, and do the data support the conclusions? The manuscript must describe methodologically and ethically rigorous research with conclusions that are appropriately drawn based on the data presented.

Reviewer #2: Yes

3. Has the statistical analysis been performed appropriately and rigorously?

Reviewer #2: Yes

4. Have the authors made all data underlying the findings in their manuscript fully available (please refer to the Data Availability Statement at the start of the manuscript PDF file)?

Reviewer #2: Yes

5. Is the manuscript presented in an intelligible fashion and written in standard English?

Reviewer #2: Yes

6. Review Comments to the Author

Reviewer #2: The authors have addressed all the comments.

7. PLOS authors have the option to publish the peer review history of their article (what does this mean?). If published, this will include your full peer review and any attached files.

**Do you want your identity to be public for this peer review?** For information about this choice, including consent withdrawal, please see our Privacy Policy.

Reviewer #2: **Yes: **Rajat Das Gupta
